# Microfluidic Methods for Generation of Submicron Droplets: A Review

**DOI:** 10.3390/mi14030638

**Published:** 2023-03-11

**Authors:** Biao Huang, Huiying Xie, Zhenzhen Li

**Affiliations:** School of Aerospace Engineering, Beijing Institute of Technology, 5# ZhongGuanCunNan Street, Haidian District, Beijing 100081, China

**Keywords:** submicron droplets, nanoemulsions, microfluidics, tipstreaming, electrospray, step-emulsification

## Abstract

Submicron droplets are ubiquitous in nature and widely applied in fields such as biomedical diagnosis and therapy, oil recovery and energy conversion, among others. The submicron droplets are kinetically stable, their submicron size endows them with good mobility in highly constricted pathways, and the high surface-to-volume ratio allows effective loading of chemical components at the interface and good heat transfer performance. Conventional generation technology of submicron droplets in bulk involves high energy input, or relies on chemical energy released from the system. Microfluidic methods are widely used to generate highly monodispersed micron-sized or bigger droplets, while downsizing to the order of 100 nm was thought to be challenging because of sophisticated nanofabrication. In this review, we summarize the microfluidic methods that are promising for the generation of submicron droplets, with an emphasize on the device fabrication, operational condition, and resultant droplet size. Microfluidics offer a relatively energy-efficient and versatile tool for the generation of highly monodisperse submicron droplets.

## 1. Introduction

Emulsions stand for the complex fluids with droplets dispersed in continuous phase. Submicron droplets by definition are droplets with size in the order of 100 nm, which are equivalent to the “nanoemulsions” defined in previous reviews [1,2]. This size range is in contrast with “macroemulsions”, which stand for emulsions with droplet size in the order of 1 μm or bigger, and “microemulsions” for those in the order of 10 nm. The size of the submicron droplets endows them with privileged capacities, including mobility in confined geometries with submicron size [3]. In biomedical applications, the submicron droplets with properly coated surfaces are able to escape from the detection of the immune system while moving in blood vessels [4]. They can be used as contrast agents in cancer tissues for imaging diagnosis, and as carriers for drug delivery [5]. In the energy industry for the enhanced oil recovery, especially in rocks with characteristic pore size at submicron scale, submicron droplets are used as carriers of surfactants that help to emulsify and recover oil [6].

Another advantage of the submicron droplets relies on their high dispersity and large surface-to-volume ratio reaching up to 10 m^2^/cm^3^ [2]. This allows for submicron droplets that are widely presented in fields such as food [7], cosmetics [8], drug delivery [9], and composite materials [10]. Recently, in photo harvesting materials, liquid metal (LM) submicron droplets as the thermal conduction elements in a composite material, contribute to solar-heat-electric conversion [11]. In therapeutics, drugs can be loaded into the shells of core-shell submicron droplets [12,13], where an increased surface area is beneficial for drug-loading efficiency.

Another remarkable feature that differentiates the submicron droplets from their macro counterparts is stability. The macroemulsions are both thermodynamically and kinetically unstable. Submicron droplets are thermodynamically unstable; however, they are kinetically stable over a long-time scale [2]. Several mechanisms of instability for the emulsions have been concretely described in reviews [1,2] and we summarize them as follows. Firstly, due to density differences between the dispersed and the continuous phases, the droplets—especially those with micron size—sediment or float up (i.e., creaming), leading to a heterogenous medium. The advantage of submicron droplets is that they are prone to Brownian motion and are able to overcome the effect of gravity. Secondly, the stability of emulsions largely relies on the physical and chemical properties of the surfactants. The ionic surfactants may prevent the flocculation of the emulsion droplets by electrostatic repulsions, and the non-ionic surfactants realize the same function by steric effects. By optimization of the surfactant choice and the emulsifying procedures, multi-emulsions with complex morphology can be formed [2]. Thirdly, the coalescence between the droplets occurs if the thin film between two droplets drains, and this effect can be delayed for submicron droplets because of their increasing non-deformability due to high Laplace pressure. Fourthly, the Ostwald ripening, which consists in the transfer of the disperse fluid from smaller droplet into larger ones, caused by higher Laplace pressure in the smaller droplets, is a dominating mechanism of instability for submicron droplets. The disperse phase coming out of the smaller droplets diffuses through the continuous phase until merging with the bigger droplets, therefore, the solubility and diffusivity of disperse phase in continuous phase affects the ripening rate. The law of ripening kinetics is predicted by Lifshitz and Slyosov [14], and the important variables that regulate the instability through chemical component, concentration, and temperature are reviewed by Gupta et al. [1]. Based on the nature of the Ostwald ripening, it would be preferable to have an even distribution of droplet size at the initial stage, for the purpose of remaining as a stable nanoemulsion.

The microfluidic method is a promising technology in the formation of droplets and bubbles [15,16], with accurately controlled size and monodispersity that is superior to alternative methods. The formed droplets and bubbles are mostly suitable for biomedical applications and synthesis of materials with complex components and structures [17]. Passive and active methods were both exploited to achieve droplet generation [18]. Passive methods consist in the formation of droplets merely based on channel confinement and capillary instability, the droplets are formed in channel geometries of T-junction, flow-focusing, co-flow, step-emulsification, among others. The droplet sizes in most of the methods are imposed to be at the same order of channel width and depth, and can be regulated by variation of fluid properties and flow rates. The mechanisms of fluid thread breakup are extensively reviewed by Stone [19] and Anna [20]. On the other hand, the active methods draw support from the external energy inputs such as electrical, magnetic, sound, and mechanical forces [21], that are able to trigger fluid thread breakup if pure hydrodynamic effects (i.e., viscous and inertial effects) are insufficient.

The microfluidic methods are generally believed to form droplets at micron scale or bigger, while the formation of submicron droplets is hindered by the technical limit on nanofabrication, more specifically on the resolution of channel width and depth. However, several microfluidic methods still show promising performance in the generation of submicron droplets, such as hydrodynamic tipstreaming, electrospray, and step-emulsification. This review is aimed at conveying the potential of microfluidic technology in generating highly monodisperse submicron droplets, in contrast to bulk methods that may involve large energy input or rely significantly on chemical components. The physical origin, operational condition, and the determination of the droplet size will be discussed. Besides, we summarize the emerging applications where monodisperse submicron droplets may be crucially important. In the end, recently developed channel fabrication technology that may contribute to the nanodroplet generation are presented.

## 2. Methods of Submicron Droplets Generation

### 2.1. Bulk Methods

We refer to bulk methods as those processed in bulk within containers. These include the high-energy methods such as widely used high-pressure homogenization and sonication, with high input energy density ε that reaches $10^{8}–10^{10}\;W/\mathrm{kg}$, necessary to overcome the surface energy barrier for submicron droplet formation. The low-energy methods that require external energy input at the order of $10^{3}–10^{5}\;W/\mathrm{kg}$ can also be processed in bulk. These methods are reviewed in [1,22] with emphasis on mechanism and chemicals involved; here we briefly mention them in contrast to microfluidic methods.

#### 2.1.1. High-Energy Methods

High-pressure homogenization (HPH) forms submicron droplets by forcing the macro-droplets through a channel with high pressure, as sketched in [1]. The procedure can be repeated until submicron droplets with satisfying size and uniformity are obtained. Sonication methods lead to the formation of cavitation within the medium, the collapse of cavities produces shock waves in the fluid that break the macro-droplets into submicron droplets. The sonication provides good conditions for mixing of the components and self-assembly of the molecules. As reviewed by Zhang et al. [22] on the formation of drug-loaded PFC ultrasonic contrast agents, both drug loading and formation of lipid-stabilized PFC submicron droplets are promoted by sonication.

Both high-pressure homogenization and sonication consists in providing the medium with a strong elongational and shear stress. Gupta et al. [23] provided a critical condition above which a parent droplet can be extended and break into submicron droplets with diameter d: $We\sim Oh^{0.4}$, with $We=\frac{{\left(\rho_{c}\mu_{c}\varepsilon \right)}^{1/2}d}{\sigma }$ the ratio between applied stress and the surface stress, and $Oh=\frac{\mu_{d}}{{\left(\rho_{d}\sigma d\right)}^{1/2}}$ the ratio between viscous effect and surface effect that depends purely on properties of the fluids and is independent from applied external forces. $\rho_{c}$ and $\rho_{d}$ stands for density of continuous and disperse phases, respectively, $\mu_{c}$ and $\mu_{d}$ for viscosity of the two phases, and σ for the surface tension between continuous and disperse phases. This scaling law was validated for both homogenization and sonication. The high energy methods are practical in forming submicron droplets with size controlled by input energy, processing time, and fluid properties [23]. The sample quantity can be made to greatly satisfy the industrial demand. The backdrop consists in the violent agitation of the fluid medium that might bring destructive effect to fragile molecules such as polymer chains, proteins, and DNA. Since heat is severely dissipated during the application of high-energy methods, the energy efficiency is low and only about 0.1% of the input energy is used for emulsification [24].

#### 2.1.2. Low-Energy Methods

The low-energy methods in forming submicron droplets profit from thermodynamic properties of fluids and of surfactants, thus they do not require high external energy input. These low-energy methods are divided into two categories in the review of Solans and Solé [25], namely, self-emulsification and phase inversion. Self-emulsification is triggered by the addition of the continuous phase into a microemulsion (emulsion droplets of order 10 nm), that induces diffusion of components from the disperse phase to the continuous phase, the initial microemulsion is no longer thermodynamically stable and evolves into nanoemulsion (emulsions with submicron droplets) [25]. The self-emulsification does not involve phase inversion. On the other hand, submicron droplets can be formed via phase inversion, which can be triggered by the addition of a chemical component (phase inversion component—PIC) or by the variation of temperature (phase inversion temperature—PIT) [25]. In the PIT method, the mixture is firstly prepared at the temperature of phase inversion, at which the curvature of the interface is extremely low, the temperature is subsequently raised or reduced to form submicron droplets of W/O or O/W. In the PIC method, a component fluid is progressively added into the mixture, which allows inversion of the interface curvature and formation of submicron droplets. The interfacial energy is provided by the chemical energy released from the medium, so that the low-energy methods rely significantly on the chemical components that are involved.

#### 2.1.3. Bubble Bursting

Bubble bursting is a newly developed method to efficiently produce submicron droplets [26]. A thin film of oil covers the water free surface, forming an air/oil/water compound interface. Bubbles are created in the water bulk, and rise to the compound surface causing the oil film to drain, and subsequently the water film bursts, inducing a strong shear to the oil film, and a cloud of oil submicron droplets are formed and are suspended in the water. The droplet size can be regulated by the physicochemical interactions between oil molecules and surfactants, instead of the bubble dynamics.

The bulk methods have good performance in massive production of submicron droplets that can satisfy industrial demands, however, they are restricted to polydispersity of droplet size and reproducibility. The high precision of droplet size control is the strong suit of microfluidics. Although the production rate of microfluidics poses a new challenge, numerous attempts have been made to develop this technology toward generation of evenly distributed submicron droplets with improved throughput. In the next section, we will discuss three microfluidic technologies that have remarkable performance in forming submicron droplets.

### 2.2. Microfluidic Methods

The advantage of microfluidics in forming submicron droplets consists, but is not limited to, in the following points. Firstly, monodisperse droplets have significant implication in diagnosis and drug delivery. For instance, PFC submicron droplets as ultrasound contrast agents with even size distribution would provide even signals when they are subjected to the same ultrasonic wave. Secondly, microfluidic methods based on laminar flow are much more energy-efficient than high-energy methods. The energy efficiency defined as the ratio of required surface energy and energy input is estimated to be 50–60% in a membrane emulsification microfluidic device [27]. In addition, the elongational and shear effects are more friendly to fragile molecules than high-energy homogenization and sonication methods. Thirdly, the hydrodynamic method of microfluidics is less selective on chemical components than the low-energy methods, although surfactants or additives are still required to be used for the proper formation and stability of the droplets. Microfluidics has been extensively developed in the past decades, the methods and theories have been presented in excellent reviews [17,18,20], while here we focus on the methods which are specifically performant in generating submicron droplets, and not very exigent on sophisticated nanofabrication. They are hydrodynamic tipstreaming, electrospray, and step-emulsification.

#### 2.2.1. Hydrodynamic Tipstreaming

Among the microfluidic methods, disperse fluid can be most effectively emulsified into droplets and bubbles by the geometries of flow focusing [28,29] and co-flow [30]. Device geometries and mechanisms of the two methods are thoroughly described in reviews [18,20,31]. In the latter geometry, the disperse phase—once coming out from the supplying capillary—flows in parallel with the continuous phase, which exerts a tangential viscous shear stress. Whereas the flow focusing geometry possesses an orifice at a distance downstream of the feeding capillary. This causes a pressure gradient through the orifice, and the dispersed fluid is stretched streamwise due to shear and elongational effect from the continuous phase [31]. According to different flow conditions, dripping and jetting modes occur, which differ from one another on the length of thread at the first pinch. The dripping mode forms droplets near the nozzle (for co-flow) or orifice (for flow focusing), and the thread breakup results from an absolute instability. Whereas the jetting mode generates droplets at a distance further downstream, and the droplet pinching results from a convective instability [20]. The dimensionless numbers characterizing these flow conditions are the Reynold number Re=ρUR/μ signifying the ratio between inertial and viscous effects, the Weber number $We=\rho U^{2}R/\sigma$ as the relative importance between hydrodynamic pressure with the Laplace pressure, and the capillary number Ca=μU/σ comparing the viscous effect with the interfacial effect, with ρ signifying fluid density, U the characteristic velocity, μ the viscosity, R the characteristic length scale, which is usually the orifice or nozzle diameter, and σ the surface tension. In flow focusing geometry, the orifice provides an extra focusing effect to the disperse phase, in contrast to the co-flow geometry. Thus, in the same flow condition, the flow focusing causes a transition from dripping to jetting modes at a smaller Ca than that in co-flow, meanwhile, this leads to smaller droplet size and higher production rate [32,33], which are more preferable for nanodroplet generation.

The axisymmetric flow focusing method proposed by Gañán-Calvo et al. is effective to emulsify the disperse phase into tiny droplets (Figure 1a), with the disperse and continuous phases being either gas or liquid [28,29]. Anna et al. [34] implemented the flow focusing strategy into a planar geometry of the microfluidic channel. A particular mode—named tipstreaming—delivering massive tiny droplets with high rate is observed, droplet diameter can be an order-of-magnitude smaller than the orifice size. The tiny droplets can be generated either in dripping or in jetting modes (see Figure 1b). The tipstreaming mode is characterized by a conical interface, from which the disperse phase can be stretched into micron- and submicron-thinned thread, and the minimum thickness of the jet can decrease down to the continuum limit [35]. This offers the microfluidics an opportunity for generating highly monodisperse submicron droplets at high throughput. Furthermore, double emulsions can be formed in tipstreaming mode using an axisymmetric flow focusing geometry, a compound thread in which the inner disperse fluid and the outer disperse fluid flow coaxially, and is focused by the air stream through the orifice [36] (Figure 1c). Circulation cells appear within the cone, when the injected disperse phase cannot be exhausted by the jet, and it is fed by the tangential surface stresses [37,38]. Numerical simulation reveals that the flow pattern within the cone can be regulated by flow condition, that an increasing flow rate of disperse phase leads to a smaller recirculation zone [36]. The stable conical-shaped interface issuing a jet can be obtained in different circumstances such as the above-mentioned flow focusing, co-flow, and selective withdrawal [39]. Tseng and Prosperetti [40] proposed a general understanding of these phenomena, that they share a common instability, when the streamlines are converged to the cone tip which has zero-vorticity.

The formation of a steady conical interface is a prerequisite for the continuous generation of monodisperse droplets in tipstreaming. In the experiments of Anna et al. in flow focusing geometry [34], the tipstreaming mode occurs at a relatively large flow rate of the disperse fluid, and at a high flowrate ratio between the focusing and focused fluids. High viscous stress is required to maintain a stable conical interface. Suryo et al. [38] investigated the operational condition in terms of the effect of flow rates and viscosity to the tipstreaming in a co-flow geometry, and proposed a scaling law for the transition between dripping and tipstreaming, that $Ca^{-1}<mQ_{r}$, where Ca stands for capillary number of the disperse fluid, and m and $Q_{r}$ for the viscosity ratio and flow rate ratio between the continuous and disperse fluids, respectively. In other words, for stabilizing the conical interface at a fixed Ca, an increase in the viscosity ratio can lower the requirement on the flow rate ratio. Parametrical numerical study shows that a stable cone is promoted for larger flowrate and larger viscosity of the focusing fluid, whereas an unstable cone occurs for larger surface tension and larger orifice diameter [36].

Gañán-Calvo and Montanéro [37] theoretically reviewed the operational conditions for obtaining a steady conical interface in a jetting mode of flow focusing. Firstly, at the initial stage when the cone is establishing, the tangential shear stress exerted by the focusing fluid should be high enough to overcome the surface stress that scales with σ/D, with σ being the surface tension and D being the diameter of the orifice. Secondly, more energy is needed for maintaining the jet in which the focused fluid is accelerated, the Weber number We>1, meaning physically that the kinetic energy acquired by the focused fluid should compensate the surface energy of the jet. The operational conditions for the cone-jet structure by flow focusing are illustrated in Figure 2a.

Once a jet is formed, the jet thickness has direct influence on the droplet size. Castro-Hernández et al. [30] provided an expression of droplet size using measurable parameters of the jet, including the jet diameter $d_{j}$, the flow rate of the focused fluid Q, the velocity of the tip $U_{p}$, and the wavenumber $k^{*}$ of the perturbation with maximum growth rate. Irrespective of how the jet is formed, the droplet diameter is given as $d_{d}={\left(\frac{6Qd_{j}}{k^{*}U_{p}}\right)}^{1/3}$. In the case of a thin jet issuing from tipstreaming, $\frac{Q}{U_{p}}$ scales with $d_{j}^{2}$, so that the droplet size is proportional to the steady jet diameter. Furthermore, the scaling laws of jet diameter based on driving factors are deduced for flow focused by gas [29,37]. The pressure difference $\Delta P_{g}$ applied to the gas is converted into kinetic energy of the focused fluid, yielding the diameter of a steady jet $d_{j}$ as function of the flow rate Q and $\Delta P_{g}$:(1)$$d_{j}={\left(\frac{\rho Q^{2}}{2\pi^{2}\Delta P_{g}}\right)}^{1/4}$$

Note here that the fluid properties such as viscosity and surface tension are not present. Equation (1) is consistent with numerous experimental results on the normalized droplet radius that $R_{d}/R_{\sigma }\sim We$, where $R_{\sigma }=\sigma /\Delta P_{g}$ [37] (see Figure 2b).

The axisymmetric flow focusing and co-flow geometry for tipstreaming can also be realized in a planar PDMS channel. To avoid the instability of the cone-jet in 2D geometry, 3D elements are introduced into the PDMS channel. For instance, a PDMS block with step geometry is sealed with its PDMS mirror block [41], so as to pave the way for the steady axisymmetric conical shape and deliver submicron droplets. An alternative method is to insert two glass capillaries with flame-shaped tips into the planar PDMS channel, so as to form a hybrid device with a 3D orifice [42]; submicron droplets of perfluorocarbon (PFC) fluid can be produced for the ultrasound contrast enhancement (Figure 3a). In addition, massive production of PFC submicron bubbles stabilized by lipids can be realized by a PDMS flow-focusing device at throughput of $10^{6}$ bubbles per second [43] (Figure 3b).

Alternative experimental methods are proposed to reduce thread diameter and droplet size to submicron scale. The focused fluid may flow around the external surface of a needle with pointed end [44], forcing the formation of a conical interface and consequently a thin jet. In addition, emulsion droplet size can be tuned by changing relative position between the needle tip and the orifice, while maintaining the same flow rates. Another method allowing to explore the minimum jet thickness is based on an opposed flow focusing geometry, it creates a back flow which brings the disperse thread to tipstreaming mode [45]. The thread can be reduced to submicron thickness by increasing the flow rate of the focusing fluid and decreasing that of the focused fluid, without transition to the dripping mode, since the continuous phase flow issued from the opposed direction does not provide a focusing effect (Figure 3c). Recently, a rotary-flow-shearing strategy was realized with two counter-rotational cylinders, which provide strong shear force to the disperse fluid issued from a capillary [46] (Figure 3d). Non-spherical liquid metal particles can be generated. This method has the potential to be extended to form submicron droplets at suitable flow condition and fluid property.

The tipstreaming mode in hydrodynamic focusing or co-flow usually requires high shear stresses provided by the continuous phase. Once the outer shear stress is not high enough, alternating driving forces such as electrical force may be used to achieve a similar steady conical meniscus issuing the submicron droplets. This is referred to as the electrospray technique, which has received extensive experimental and theoretical study [47].

**Figure 3 micromachines-14-00638-f003:**
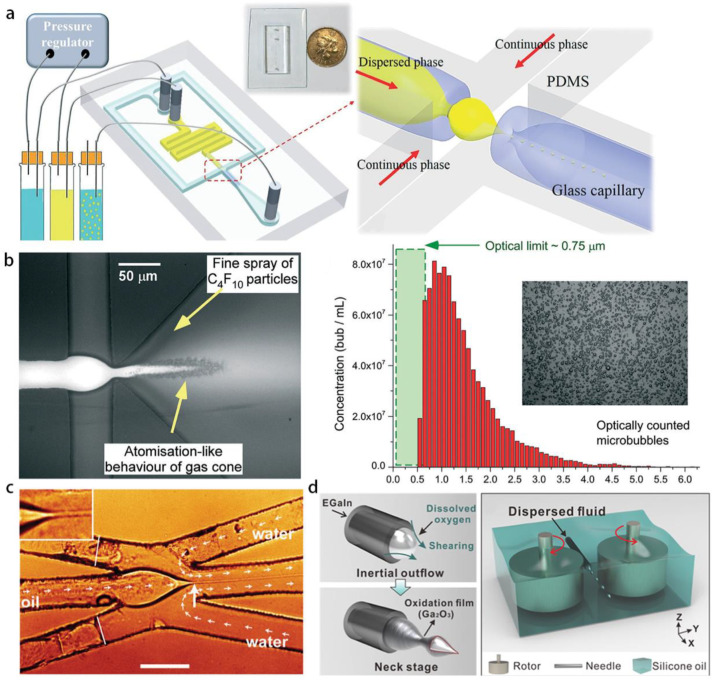
Recently developed microfluidic strategy with high emulsifying performance for tiny droplets. (**a**) Flame-shaped capillary hybrid with PDMS channel for realizing 3D flow focusing, producing submicron PFC droplets. Reproduced from [42], copyright 2017, Royal Society of Chemistry. (**b**) Atomization of lipid-stabilized bubbles in a PDMS channel with 3D expansion geometry, reproduced from [43], copyright 2016, Royal Society of Chemistry. (**c**) Continuous phase back-flow induces tipstreaming with very thin jet without transition to the dripping mode, by avoiding the focusing effect, reproduced from [45], copyright 2018, Royal Society of Chemistry. (**d**) Rotary-flow-shearing method providing strong shear and elongational stress to the disperse fluid, reproduced from [46], copyright 2021, American Chemical Society.

#### 2.2.2. Electrospray

The electrospray consists in the emission of tiny droplets from a charged interface between a conductive medium and a dielectric medium [48,49]. Experimentally, the electrospray is realized by injecting the disperse phase from a metallic capillary into a dielectric medium, and applying a high voltage difference at the order of kilovolts between the capillary and a grounded electrode placed downstream of the capillary outlet [50] (Figure 4a). At properly regulated control parameters and for suitable fluid properties, the meniscus can be deformed into a conical shape known as the Taylor cone [51], a thin jet can be emitted from the tip, and a cloud of charged droplets at submicron scale can be generated [48]. The Taylor number (or the electric Bond number) $\Gamma =\frac{R_{j}\varepsilon_{0}E^{2}}{4\pi \sigma }$ describing the level of electrification of the jet, is the relative importance between the electrical stress normal to the free surface and the Laplace pressure, where $R_{j}$ stands for the jet radius, $\varepsilon_{0}$ the permittivity of the vacuum, E is the electrical field, and σ the surface tension. Next, we state the operational conditions and the jet diameter from a practitioner point of view. Readers interested in the development of the theory are referred to the extensive and profound reviews by Gañán-Calvo et al. [52] and of Rosell-Llompart et al. [53].

The steady cone-jet mode with the Taylor cone anchored to the feeding capillary has drawn lots of attention because of its importance in realizing continuous and robust generation of submicron droplets. At a given fluid property and geometry, the cone-jet stability relies on flow rate, and the applied voltage. It is required that We>1 [37] (in analogy with the hydrodynamic tipstreaming), as the kinetic energy converted from electrical potential difference overcomes the surface energy. However, unlike in hydrodynamic tipstreaming, the shear stress in electrospray plays a subdominant role, in comparison with the electrical force.

Practically, a necessary condition for having a steady cone-jet is that both the flow rate of the disperse phase and the applied electrical potential should be simultaneously confined in a range, as demarcated in the phase diagrams by Bober et al. [54]. Consider $Q_{min}$ as the minimum flow rate, below which the stable cone-jet is unable to form for any applied voltage V, and there is intermittent emission of jets. $Q_{min}$ is a function of fluid parameters (i.e., density, viscosity, surface tension, conductivity, permittivity of vacuum and of the fluid), its scaling law is deduced from a force balance, between the accelerating electrical force and the opposing forces, being the polarization force or the viscous force, as deduced by Gañán-Calvo et al. [44]. For $Q>Q_{min}$, there is an interval of V for having steady cone-jet flow, the lower boundary guarantees a sustainable cone with elongated shape, and the upper boundary corresponds to the limit where electro-hydrostatic solution no longer exists and the jet tilts laterally (Figure 4b) [52], whipping instability appears at higher Taylor number. At a fixed voltage, a finite range of flow rate delimitates the steady cone-jet mode. Detailed operational window and scaling expressions are rationalized [52,53] that are consistent with experimental observation and numerical results [55]. Study of the minimum flow rate has implications in exploring the smallest droplet size at fixed properties of fluids, and a deep understanding of the instability mechanism of minimum flow rate would propose a strategy to suppress it for obtaining thinner jets. For a tunable geometry, a feeding capillary size not sufficiently larger than the axial cone-jet transition length has a stabilizing effect to the cone-jet.

The governing forces in the cone-jet electrospray includes the axial component of the tangential electrostatic surface stress and of the normal electrostatic surface stress, and the polarization force as the three driving terms; and the surface tension force, the liquid inertia, and the viscous force in the axial direction as the three resistance terms [56]. The distribution of four principal regimes is presented in a phase diagram, with the inertial-electrical (IE) regime as the dominating one found in the experiments (Figure 4c). In the IE regime, Gañán-Calvo proposed a scaling law for the electrical current I convected by the jet and the jet diameter $d_{j}$ as:(2)$$I\sim {\left(\sigma KQ\right)}^{1/2},\;d_{j}\sim {\left(\frac{\rho \varepsilon_{0}Q^{3}}{\sigma K}\right)}^{1/6}$$
where σ is the surface tension, K, Q, and ρ are the conductivity, flow rate, and density of the liquid, respectively, and $\varepsilon_{0}$ is the the permittivity of the vacuum. Low flow rate and high conductivity and surface tension are therefore desired for having thin jets. Since the electrospray has physical analogy with the hydrodynamic flow focusing tipstreaming, but differs in driving force with the latter [37], the jet thickness $d_{j}$ in electrospray the IE regime also satisfies Equation (1), which was developed for flow focusing, by considering the electrical force induces an effective pressure difference $\Delta P_{e}=k_{p}{\left(\frac{\sigma^{2}K^{2}\rho }{\varepsilon_{0}^{2}}\right)}^{1/3}$, (in analogy with $\Delta P_{g}$ for gas pressure in Equation (1)), here $k_{p}$ is a constant at order unity. The scaling of the jet diameter in Equation (2) is compared with experimental measurements and shows unequivocal agreement, as reviewed by Gañán-Calvo [52] (Figure 4d).

**Figure 4 micromachines-14-00638-f004:**
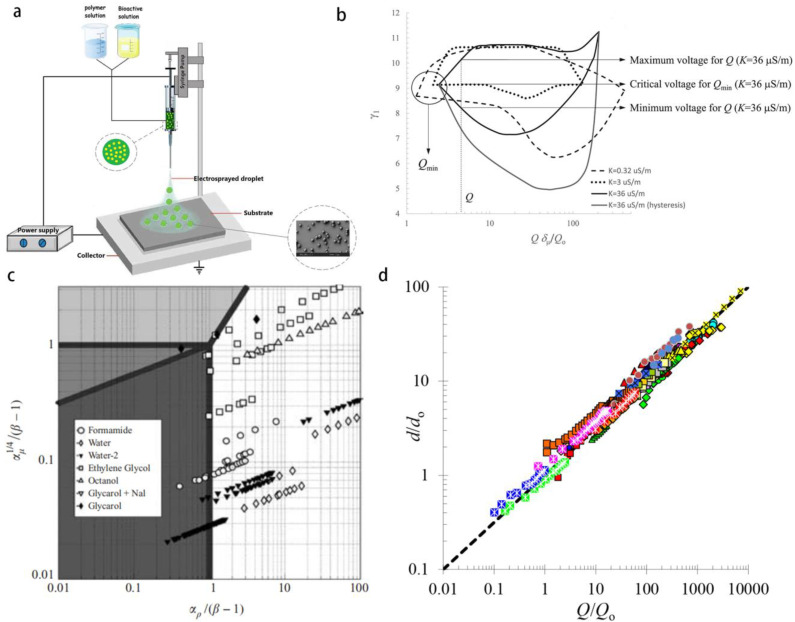
(**a**) Setup of electrospray, a potential difference is applied between the disperse phase in the feeding capillary and the grounded plate downstream, reproduced from [50], copyright 2021, Elsevier. (**b**) The operational conditions of electrospray for having steady cone-jet mode, in plane of Taylor number and normalized flow rate of disperse fluid, reproduced from [52], copyright 2018, Elsevier. (**c**) Phase diagram of 4 main regimes with dominant forces for electrospray, including the inertia-electric mode (white), inertia-polarization mode (dark grey), viscous-electric mode (light grey), and viscous-polarization mode (medium grey). Most of the experiments step into the inertial-electric (IE) mode. Reproduced from [56], copyright 2004, Cambridge University Press. (**d**) Normalized droplets diameter vs. the normalized flow rate. Here $Q_{o}=\sigma \varepsilon_{0}/\rho K$, $d_{o}={\left[\sigma \varepsilon_{0}^{2}/\left(\rho K^{2}\right)\right]}^{1/3}$, with σ the surface tension, $\varepsilon_{0}$ the permittivity of vacuum, and ρ and K the fluid density and conductivity, respectively. The power law scaling is consistent with Equation (2), and with the experiments using different fluids represented by different colors. Reproduced from [52], copyright 2018, Elsevier Ltd.

Besides direct current actuation, an alternating current (AC)-induced electrospray was developed in recent years [57]. In addition to the voltage V, the AC frequency f regulates the shape and dynamics of jet emission, different regimes are described in a (V, f)-plane, including the tipstreaming mode, elongated fast dripping mode, oscillating mode, and conical meniscus mode, respectively, from low to high frequency. A square-wave AC-actuated tipstreaming is used for generating capsules [58], at a smaller voltage in comparison with DC actuation. A stable cone jet can be obtained at a higher f than the resonant frequency of the cone that is defined by the Rayleigh–Lamb dispersion relation [57].

Electrospray was widely adopted in mass spectrometry (MS) of large molecules, because it allows a fast and efficient ionization of the analytes. In one of the pioneering works [59], the analyte droplets were sprayed out via a metallized glass capillary needle under the application of voltage, a tiny amount (order of μl) of peptide mixture is then subjected to MS. Improvements of the electrospray for MS have been made in recent years, which have extended the range of analyzable substances to nonpolar molecules and large biopolymers [60]. Practical parameters regulating the droplet size are summarized by Chakraborty [61]; the choice of polymer molecular weight and concentration affects the fluid viscosity, conductivity, and surface tension, so as to modulate the jet diameter and droplet size according to Equation (2).

Similar to other microfluidic technologies, the electrospray is promising for forming Janus droplets [62] and core-shell droplets [50] with perfect encapsulation rate. The Janus droplets are made by applying potential difference to the co-flowing disperse phases (Figure 5a), whereas the core-shell droplets can be formed by the electrospray of co-axial threads of the core and shell liquids [50] (Figure 5b), or by the separate electrospray of two oppositely charged droplets, which then impact and merge with each other under suitable condition of surface tension [47]. The core-shell structure is demonstrated to be an important template for drug delivery with controlled release [63]. The key advantage of the electrospray in drug encapsulation compared with other microfluidic methods consists in the generation of submicron-sized droplets.

Owing to the ability of emitting colloidal particles, the electrospray was also applied as the colloid thruster in propellant technology to achieve accurate positioning of small satellites. The thrust reaches the order of 0.3 μN for conductive liquid [64]. The emitters (nozzles) can be microfabricated in an array on a silicon wafer using deep reactive ion etching (DRIE) technology [65] (Figure 5c).

When the electrical field is applied to polymer solutions which possess viscoelastic properties, the jet may not break into the droplets due to high extensional viscosity, instead, a fiber ejection occurs. The nano-composites formed by the combination of electrospun polymer and TiO_2_ nanoparticles with the anti-microbial function are an emerging material for tissue engineering and wound healing [66]; the porous media formed by scaffolds of the polymer nanofibers is permeable to the air while isolating the wound place from infectious agents. The nanofibers are also used as a drug deliverer with controlled release [61]. Recently, the porosity of an electrospun polymer matrix was used as a template for nanochannel fabrication, for the generation of submicron droplets at unprecedented high throughput [67]. The matrix of electrospun fibers of polycaprolactone (PCL) is embedded into a PDMS channel with inlets for disperse and continuous phases. The microfibers are then eliminated by dissolution in dichloromethane, leaving the PDMS chip with interconnected nano-pores. The disperse phase is emulsified into submicron droplets at numerous junctions among the fibrous channels (known as the T-junction or Y-junction emulsification mode), thus this can realize a throughput in the order of $10^{10}$ submicron droplets per second (Figure 5d), which is advantageous compared with conventional microfluidic methods based on sophisticated channel nanofabrication.

**Figure 5 micromachines-14-00638-f005:**
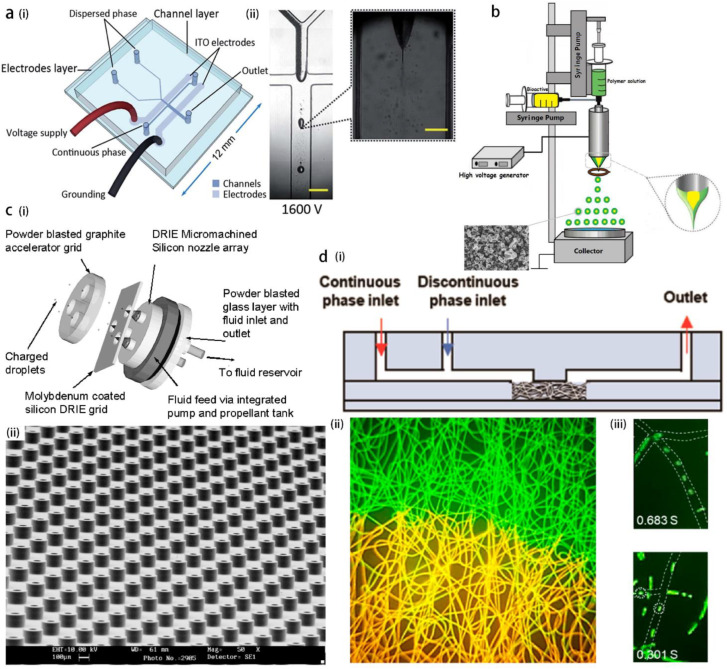
(**a**) Electrospray of co-flowing bi-disperse phases for the formation of Janus droplets, the sketch of the device (**i**) and the experimental image (**ii**). Reproduced from [62], copyright 2011, Royal Society of Chemistry. (**b**) The device for electrospray of co-axial fluids for the formation of core-shell droplets, reproduced from [50], copyright 2021, Elsevier. (**c**) Electrospray device as the colloidal thruster for accurate position regulation of aero-crafts, the device (**i**) and the arrays of silicon nozzles (**ii**). Reproduced from [65], copyright 2006, AIAA. (**d**) Electrospun nanofibers as the template of a porous media for submicron droplets generation. The matrix made from the electrospun fibers is inserted into a PDMS channel (**i**), both continuous and disperse phases flow through the porous media, with their interface shown as the boundary between yellow and green (**ii**), submicron droplets of disperse phase are formed in the pores via T-junction and Y-junction modes (**iii**). Reproduced from [67], copyright 2022, American Chemical Society.

#### 2.2.3. Surfactant Mediated Tipstreaming

Surfactant molecules are often used to reduce interfacial energy, which results in smaller droplet size and stable emulsions. The presence of surfactant also assists the droplet formation in tipstreaming mode by the reduction of local surface tension. As studied by de Bruijn [68], tipstreaming occurs at an appropriate flow condition and especially at a moderate concentration of surfactants, which creates a surface tension gradient along the interface. The numerical work of Eggleton et al. [69] reveals that within the surfactant concentration range that allows tipstreaming, a relative lower surfactant concentration leads to thinner thread, whereas thicker thread occurs for higher surfactant concentrations. It is believed that the tipstreaming is induced by convection of surfactants on the interface towards the pole of the disperse thread. The surface tension at the pole is strongly reduced by local accumulation of surfactants, a thin thread can then be drawn out from the tip [19,20]. Note that tipstreaming also occurs in surfactant-free situations, however, the surfactants facilitate the occurrence of the tipstreaming. As shown by the numerical work of Booty et al. [70], with the presence of surfactants, tipstreaming can be triggered from a slender bubble at a smaller capillary number than if no surfactant is present.

The operational condition of tipstreaming in terms of surfactant concentration and flow condition is crucial for having steady and continuous nanodroplet generation. Moyle et al. [71] theoretically constructed a phase diagram in plane of surfactant bulk concentration C and the focused fluid flow rate Q, where the tipstreaming mode occurs in a limited region. The theory is based on an argument that bulk surfactants should adsorb fast enough onto the interface, in order to realize a stress profile along the interface, which is required for maintaining a conical shape under a strong convective flow. In addition, requirements based on global surfactant mass balance, disperse phase mass balance, and minimum interfacial tension at surfactant saturation, all together delimitate an operating region for the tipstreaming. The theoretical phase diagram agrees with the experimental observation (Figure 6a), this demonstrates the importance of surfactant adsorption–desorption mechanism on the occurrence of tipstreaming, and thus on the droplet size. The Biot number (Bi) defines the relative importance between the characteristic advection time and the surfactant desorption kinetic characteristic time. The numerical simulation of Wrobel et al. [72] reveals that a larger Bi, meaning larger desorption rate from the interface, causes a shorter thread at first pinch, which is induced by higher surface tension. The shape of the interface and the interfacial surfactant distribution is calculated [72], and an abrupt increase in surfactant concentration is found on the thin thread emitted from the cone (Figure 6b).

The accumulation of surfactants on the thin thread is contrasted by the depletion of surfactants on the conical interface, where surface tension is higher. Therefore, a primary larger droplet at the scale of the orifice may firstly form, followed by tipstreaming with an elongated thin thread [73]. This intermittent process is not favorable for constant generation of submicron droplets. As a solution, an active feedback control loop is developed to constantly regulate the flow rate of the disperse fluid, based on the requirement that the conical tip is maintained at a stationary position [74].

Practically, surfactants can be added directly to the fluids, or be fabricated via chemical reaction at the interface, such as NaOH and linoleic acid at water–oil interface produces an anionic surfactant [75]. This method is applied to the microfluidic flow focusing [76]. The reaction kinetics regulate the surfactant concentration so that it is a moderate value required by tipstreaming mode. A dimensionless Damkohler number, which is a ratio between reaction rate and droplet formation frequency, is found to be in the order of O(1) for having enough surfactants synthesized before being convected to the drop tip.

#### 2.2.4. Step-Emulsification

In recent years, step-emulsification (SE) as a droplet generation microfluidic method has emerged. The pinching of the disperse thread occurs at an abrupt expansion of channel width and height, due to the capillary force. To the contrary, with the T-junction and flow focusing methods, the droplet size in the SE method has minor dependence on shear effect of the continuous phase and therefore, is less sensitive to fluctuation of flowrates. In addition, the droplet production rate is advantageous, reaching the order of 100 droplet/sec per single channel for a single disperse phase flowing in the channel [77], and $10^{4}$ droplets/sec for disperse-continuous phase co-flow in the channel. The throughput can be further increased by parallelized channels. A systematic introduction of the SE method can be found in the review articles [78,79]. In this review, we focus on the SE method, which is dedicated to downsizing of the droplet formation, and has an emphasis on the development of the mechanisms.

The key parameters to achieve generation of droplets in step-emulsification include: 1. The aspect ratio of the Hele-Shaw (H-S) channel which is required to be β=w/h>2.6. Above this limit, a backflow associated with an adverse pressure gradient is facilitated from the reservoir into the H-S channel [80,81], and the backflow promotes pinching; below the limit of β=2.6, the drop expands isotropically, that is not beneficial for pinching [81] (Figure 7a). 2. The dewetting state of the disperse phase on the channel wall, with a contact angle above 120° [82], causing the Laplace pressure in the H-S channel, which sets the condition for the capillary pinching (or necking) at the step. 3. Capillary number being smaller than a critical value $Ca^{*}$ for which the droplet production mode transits from dripping (i.e., forming small droplets at high rate) to jetting (i.e., forming bulbs at low rate) [82,83,84]. Above $Ca^{*}$, dynamic pressure inhibits the development of the instability as shown by a direct numerical simulation, and large balloons are formed [85]. The synergistic effect of disperse fluid contact angle and critical capillary number are shown in the phase diagram of Figure 7b.

The geometry of the Hele-Shaw (shallow) channel evolved from a simple straight channel to a channel with a trumpet shape [86] in connection with the reservoir. The trumpet shape promotes the backflow from the reservoir into the channel to facilitate pinching of the disperse thread. In addition, channel constriction is designed to localize the instability, and a bypass channel connecting the reservoir with the main channel profits from the adverse pressure gradient, thus increasing the speed of breakup [87]. In the geometrical design of the parallelized step-emulsification devices, there are several milestone geometries, which are dedicated to small droplet formation at high throughput: 1. Membrane emulsification can be achieved through holes with oblong shape on a silicon plate [27,88], the energy efficiency reaches 60%, micron-sized droplets can be generated from a channel with depth of three times thinner (Figure 8a); 2. Array of straight channels aligned in millipede arrangement is another efficient design for high production rate of droplets (Figure 8b). Various fluid injection methods are used, such as pressure and flow rate control [86], centrifugation [89], and buoyancy [90]. Droplet size at micron scale can be achieved by reducing the channel height to micron [77]; 3. Edge-based droplet generation method (EDGE) consists in forming droplets at the intersection of the reservoir and a plateau, which has an ultrahigh aspect ratio (e.g., 500 μm width and 1.2 μm height). Several droplets can be simultaneously formed along the plateau spanwise [91,92]. In addition, such plateau structures can be connected in series to further increase the throughput (Figure 8c). The partitioned EDGE device consists in adding equal spaced partitions at the plateau–reservoir intersection. It looks similar to the straight channel arrays, however, it avoids the hydrodynamic resistance caused by channel length [93].

One factor that may cause polydispersed droplets is the collision between the formerly formed droplets and the forming droplets. Different collision modes lead to bigger or smaller droplets. In addition, the droplet production rate of single channel fluctuates for different channel locations [77]. The conception of a bypass channel, which injects the continuous phase into the reservoir, helps clearance of the freshly formed crowded droplets, thus avoiding interference among the droplets [87].

The studies devoted to the mechanism of step-emulsification will be summarized in the following, which reveal the origins of the above characteristics of the SE method, and gave theoretical underpinning for the resulting droplet diameter and production rate.

Dangla et al. [94], based on a geometric argument, proposed a mechanism of quasi-static pinching of the dispersed thread. The interfacial curvature of the confined thread upstream of the step has to be balanced with the decreasing curvature of the growing bulb, thus inducing necking of the thread until the Rayleigh–Plateau instability takes place. The R-P instability is triggered when the width of the disperse thread decreases down to a critical value, which can be regulated by the wetting state of the disperse phase on the channel wall. The contact angle also plays a crucial role on the critical capillary number for transition from dripping to jetting mode, and on regulation of droplet size [82]. The channel height b sets the order of magnitude of the droplet diameter, which has a linear dependence on b in the dripping mode, whereas the capillary number has a minor effect [94], an increase in capillary number associated with an increasing flow rate of disperse phase rather contributes to the increasing production rate of droplets. The characteristic timescale of the breakup of the disperse thread—the necking time—is tightly related to the resulting throughput and droplet size. A semi-empirical law for the necking time and droplet size was proposed by Crestel et al. [80], accounting for the channel aspect ratio, the viscosity ratio between disperse and continuous phases, and the flow rate.

The above step-emulsification methods consist in a single disperse phase flowing in a non-wetting channel, droplets are formed at a moderate production rate per single channel, and the scaling up of the throughput is realized by parallelization of the channels. For instance, the microchannel emulsification plate forms 134.5 droplets per second in a single channel, with 176,176 microchannels in an array that achieves massive production of droplets at micron scale [88]. Besides, a derivation of the step-emulsification method that allows satisfactory production rate per single channel in forming micron and submicron droplets is the co-flow step-emulsification (cf-SE) that was first reported by Priest et al. [83]. A critical value of capillary number Ca=μU/σ associated with the width of the disperse thread in the upstream defines the boundary between two regimes, one is the step-emulsification (BE) regime in which monodispersed small droplets are formed at high frequency, and the other is the balloon-emulsification (BE) regime with large balloons formed at low frequency [83]. The disperse phase non-wetting the channel is englobed by the continuous phase, both phases co-flow in a Hele-Shaw channel until arriving at a geometrical step, where the disperse thread breaks into droplets. Li et al. [84], based on Hele-Shaw (H-S) dynamics, proposed a model describing the width variation of the disperse thread in the upstream of the step. Besides the theory with quasi-static equilibrium assumption [94], the model of H-S dynamics [84] takes into account the viscous forces and assumes an equal pressure at the exit of the step, the disperse thread accelerates due to larger pressure gradient leading to thread thinning. In this model, we defined a capillary number $Ca=12\mu_{1}q_{1}/\sigma bw$, with $\mu_{1}$ and $q_{1}$ being the viscosity and flow rate of the disperse phase, respectively, σ the surface tension, and b and w are the height and width of the Hele-Shaw channel, respectively. A critical capillary number $Ca^{*}$ sets the boundary between the SE and BE regimes. A theory concerning the capillary pinching at very low Ca explains droplet size dependence on flow rate ratio, however, the regulation of flow rate ratio is marginal, and the droplet size is principally imposed by H-S channel height. Numerical calculation of an axisymmetric step-emulsification in a problem of co-annular flow shows that droplet size is independent from Ca, which demonstrates that thread pinching in cf-SE purely originates from capillary effect [95]. The thread pinching requires a sudden expansion of the channel width or height with a large enough expansion ratio, as shown by the numerical study on a biphasic flow in a H-S channel with constant height, a channel width expansion ratio equal to three induces significant thinning of the thread near the step, however, is not large enough to cause pinching [96]. The cf-SE can also be applied to tri-phasic flows to form double emulsions with controllable shell thickness [97]. The main advantage of the cf-SE method consists in the throughput per single channel. Malloggi et al. using this method formed droplets with diameter ranging from submicron to microns [98]. A production rate of $1.5\times 10^{4}$ submicron droplets can be formed in a single channel with height h=300 nm (Figure 9a). Shui et al. [99] realized droplets ranging from 400 nm to microns using a cf-SE device with a nanoscale channel, the throughput reaches the order of $10^{5}$ per single channel (Figure 9b). The cf-SE allows single-channel high production rate of monodispersed submicron droplets with size non-sensitive to flowrates. We can further expect the cf-SE to combine with nano-device fabrication with channel arrays, using the strategy of parallelized two-phase channels [100], to optimize the submicron droplets production to over a trillion per hour.

While step-emulsification along with other conventional microfluidic methods are advantageous in generating highly monodisperse droplets with precisely controllable diameter, micron- and nano-fabrication is sophisticated and sometimes not easily accessible. In this circumstance, the microfluidic technology may firstly form droplets at micron or larger scale, which subsequently shrink under chemical or thermodynamical processes.

### 2.3. Microfluidics Combined with Low-Energy Methods

Droplets composed of low-boiling-point fluids are promising in switching size under the regulation of temperature, ultrasound, or irradiation, and the droplets can vaporize so that they expand to bigger size [22]. Reversely, the submicron droplets can be formed by condensation of gas bubbles generated in a microfluidic channel. Seo and Matssura [101] formed perfluoropentane (PFP) submicron droplets using this strategy of phase condensation. The microfluidic method guarantees the formation of monodisperse bubbles, so as to obtain submicron droplets with even size.

The disperse fluid can also be mixed with an auxiliary fluid, and the mixture is emulsified by the microfluidic device into large droplets or bubbles. The auxiliary fluid can be extracted by dissolution or evaporation, thus leaving a smaller droplet of the disperse fluid [102]. For instance, a mixture of low-saturate-vapor-pressure (LSVP) fluid and high-saturate-vapor-pressure (HSVP) fluid forms droplets suspended in oil, followed by the evaporation of the HSVP fluid, smaller droplets of LSVP fluid can be generated [103]. Furthermore, a bubble composed of a mixture of a soluble gas and an insoluble gas can shrink due to the dissolution of the soluble gas, the decreasing diameter further increases the Laplace pressure within the bubble and leads to phase condensation of the insoluble gas toward the formation of submicron droplets [104]. In other works, which are destined to form micro- and nano-bubbles, similar strategy of solvent extraction can be adopted. The bubble size can further be controlled by the maximum packing density of lipids at the gas–liquid interface [105].

In the above subsections, we have discussed different microfluid strategies in comparison with the bulk methods, for the generation of submicron droplets. Table 1 summarizes the ability of the methods in terms of droplet diameter, size distribution, and throughput. Bulk methods including high pressure homogenization, bubble bursting, and low energy methods generate droplets with diameter below 100 nm at superior throughput. Polydispersity is difficult to avoid because of lack of control at submicron level. Microfluid methods including the tipstreaming and step-emulsification greatly improve on the even size distribution, meanwhile compromises on the production rate are several orders of magnitude lower. Microfluidic method combined with low energy methods are able to make submicron droplets at satisfying throughput, however, this relies more significantly on chemical components. The newly developed technique consisting the formation of a PDMS porous media based on a solvent removable matrix of electrospun fibers is able to generate 100 nm droplets, at comparable high throughput with bulk methods, meanwhile, it provides a friendly emulsifying environment for the fragile molecules [67].

## 3. Emerging Applications of the Submicron Droplets

The nanoemulsions contribute in a wide range of fields [6,7,8,9] where the submicron droplets act as containers or carriers of the functional components. Their submicron size allows for mobility in highly confined pathways, and the high surface-to-volume ratio increases the component loading efficiency. Among the diverse fields, we focus on some newly developed applications of submicron droplets in the energetic and biomedical fields.

The submicron droplets are applied in enhanced oil recovery (EOR), which consists in the detection and recovery of oil trapped in low-porosity rocks [110]. Among all kinds of EOR techniques including thermal EOR, gas EOR, and chemical EOR [111], the latter recovers oil through the injection of chemicals such as surfactants, polymers, and alkalis [112]. The surfactants play an important role in reducing the surface tension between oil and water phases, also switching the wettability of the rock surface from oil-wet to water-wet, which is aimed at detaching the retained oil from the solid surface. The injection of surfactant aqueous solution risks adsorbing a large number of surfactants on water-wet rock surfaces before reaching the positions of retained oil. To solve this problem, submicron droplets can act as the carriers of the surfactant molecules, avoiding the unfunctional adsorption on water-wet surfaces and promoting adsorption on oil-wet surfaces [6]. In addition, the tunable rheology of the nanoemulsions renders the viscoelastic property to the flooding solution, which is able to “pull-and-drag” the trapped oil [113]. To be applicable in the rocks with submicron scale, the submicron droplets are expected to be in the order of 100 nm or smaller to avoid clogging. Moreover, submicron droplets with triggered release of substances [114] are in demand for improved detection accuracy and recovery performance. Microfluidics are expected to realize designable submicron droplets to meet these future requirements.

In the energy field, nanoemulsions are also used for thermal energy storage (TES) and transfer. Materials with high specific heat capacity are used as heater or coolant, in addition, phase change materials (PCM), which have the ability to absorb or release latent heat during phase transitions while the temperature remains almost constant, are ideal materials for thermal processing [115]. The PCM can be emulsified into submicron droplets, and takes advantage of the enhanced mobility and the high surface-to-volume ratio to achieve more efficient heat transfer [2], and specific heats can be increased [116]. The presence of submicron droplets in the material can also modify the viscosity, thermal conductivity, and surface tension. These properties—along with latent heat and specific heat capacity—can be regulated to realize better thermal storage [10]. However, the submicron scale of the droplets may induce challenges of increased supercooling, which is unexpected in the recycling of the PCM when the phase transition occurs at lower temperature than the melting point, which would cost extra energy input to reduce temperature for reaching the phase transition. This issue was addressed by the appropriate formulation of the nanoemulsions [117]. In a photo-thermal-electric material for solar energy storage, liquid metal (LM) submicron droplets are coated with dual layers including a photo-adsorbing layer and a photo-thermal conversion layer. The LM nanodroplet in the core contributes to heat conduction [11]. In light upconversion materials, which are able to absorb low-energy light and emit high energy light, the submicron droplets are used as containers of the upconversion dyes [118]. The production of highly monodisperse submicron droplets would offer better kinetic stability and functional efficiency to the materials for thermal energy storage and upconversion.

Ultrasound technology as an effective imaging technique, in which microbubbles are used as contrast enhancement agents, profits from their high echogenicity and their nonlinear acoustic response to ultrasonic excitation. When a sequence of pulses is sent with the same amplitude and the opposite phases, the linear echoes from tissues are cancelled out, and the scattered signals are composed of nonlinear echoes from the bubbles [119], which become distinguishable from the tissues. However, the microbubbles (with size at micron scale) are inappropriate in some circumstances such as cancer therapy for which the tumor endothelial space is only 100–780 nm [3]. Submicron agents are expected, and perfluorocarbon (PFC) submicron droplets are developed [120]. The PFC fluids are a set of low-boiling point fluids, which can be vaporized by energy input such as ultrasound or near-infrared (NIR) irradiation [16]. The resultant bubbles act as contrast enhancement agents for ultrasound imaging. Meanwhile, the vaporization causes mechanical forces that lead to formation of pores on cell membranes, so that the chemical substances loaded with the PFC droplets can be released into deeper tissues to achieve chemotherapy [12,13]; and the components with strong adsorption of irradiations and high photothermal conversion efficiency can be carried with the PFC to achieve photothermal therapy [121,122]. The related applications and chemicals used are reviewed in detail by Paknahad et al. [16] and Zhang et al. [22]. The size of the bubbles has a strong influence on the dynamic response of ultrasonic excitation at a given excitation frequency [123], a wide size distribution leads to lower rate of signals. In the PFC method, the bubbles usually have three to five times larger diameter of the PFC submicron droplets [124]. Therefore, it is of crucial importance to generate PFC submicron droplets with controlled size and low variance. In addition, this also allows a predictable concentration of drug loading, which may improve the effectiveness of cancer therapy.

## 4. Device Fabrication

Gañán-Calvo pioneered hydrodynamic flow focusing geometry [29], in which the disperse fluid issued from a nozzle is dragged by the co-flowing air through an orifice drilled on a metal plate (Figure 1a). The concept of the orifice was quickly developed into microfluidic technology based on glass capillaries, which are mainly manufactured by two methods: the heating-pulling and the flame-shaping [125,126]. In heating-pulling, a borosilicate capillary is drawn into a long and tapered shape, the diameter of the nozzle is less reproducible and may be fragile if it is reduced to micron size. To the contrary, the flame-shaping method shows better performance and reproducibility [126]. In the flame-shaping process, the capillary is positioned vertically and rotates on its axis, while being heated by a propane torch, and the capillary wall thickens at the heated end, leaving a tapered channel with a smoothly varying sidewall. The flame-shaped tube is inserted and aligned concentrically with a hollow-core optical fiber, which acts as the feeding capillary of the disperse fluid, thus forming a flow-focusing device for single emulsion (Figure 10a(i,ii)) [108]. Furthermore, double emulsions can be formed with co-injection of three fluids in an assembly of three capillary tubes. In particular, the continuous phase is injected from the opposite direction of the flowing disperse fluids, dragging them through the orifice (Figure 10a(iii)) [125].

Metallic components are required for the electrospray setup. The potential difference is either applied to the metallic piston pushing the conductive liquid [50], or directly applied to the metallic needle supplying the disperse fluid [127]. The grounded electrode can either be a metallic plate downstream of the spray [50], or a metallic ring [127]. A tri-phasic co-axial gathering of needles can be used to generate multi-layered droplets [128], and is a promising manner in which to form encapsulated agents at submicron scale.

The abovementioned tipstreaming devices all rely on manual fabrication and assembly of the components, for which a delicate imprecision may introduce remarkable variations on the jet performance. In addition, the geometry of the entire device is restricted to components’ shapes, so that it is difficult to optimize. Recently, Knoška et al. [129], using direct laser writing, printed a 3D micro-nozzle (Figure 10b), which allows formation of submicron jets at speeds reaching 160 m/s. It not only circumvents the labor-intense procedure of component alignment and assembling, but also greatly improves the reproducibility of device fabrication. As the sample delivery system for crystallography technology, this micro-nozzle fits within the repetition rate required by the Megahertz X-ray free-electro lasers (XFELs). Three-dimensional printing devices provide new insights into micron- and nano-fabrication for the controlled generation of miniaturized structures.

The ability of massive device production is also shared by PDMS molding technology, which plays a significant role in channel fabrication due to the ease of manipulation, low cost, and biocompatibility [130]. The PDMS is molded from a silicon wafer, which is fabricated via soft-lithography. Channels composed of several layers (usually less than three layers) can be realized by alignment of the mask. The in-plane precision of the channel is subjected to the mask printing resolution, and submicron channel width can be realized using chrome masks [131]. The channel depth can be reduced to submicron scale simply by regulating the spin-coated layer thickness of the SU-8 resin. This provides advantages to the fabrication of the step-emulsification device with droplet size relying principally on the characteristic depth of the channel [98] (Figure 9a).

However, the PDMS channel has its own drawbacks, as the channel may undergo deformation or even collapsing because of the nature of soft material, and swelling if infused by some organic solvents [132]. The silicon wafer and glass have good chemical and thermal resistant performance. Soft-lithography combined with deep reactive ion etching (DRIE) is a key technology in fabrication of highly compact channel arrays (10,260 flow focusing elements) on a four-inch silicon wafer [100]. The difficulty of channel parallelization is circumvented by separating the droplet generation channels and fluid feeding channels into different layers that are interconnected at different depths (Figure 10c). Therefore, throughput of monodisperse droplet generation is greatly increased to one trillion droplets per hour, shedding light on the industrial conversion of droplet microfluidics.

In summary, microfluidics has provided a platform for the generation of submicron droplets with precise control of the size and its distribution, profiting from the combination of micro- and nano-fabrication, fluid regulation, and additional driving forces. However, these methods still face the challenges of matching the throughput requirement from the industry. High production rate is currently difficult to achieve along with good size and morphology control. High resolution 3D printing of the nozzles is expected, and the rigid devices with paralleled channels and good replicability would provide important impetus.

**Figure 10 micromachines-14-00638-f010:**
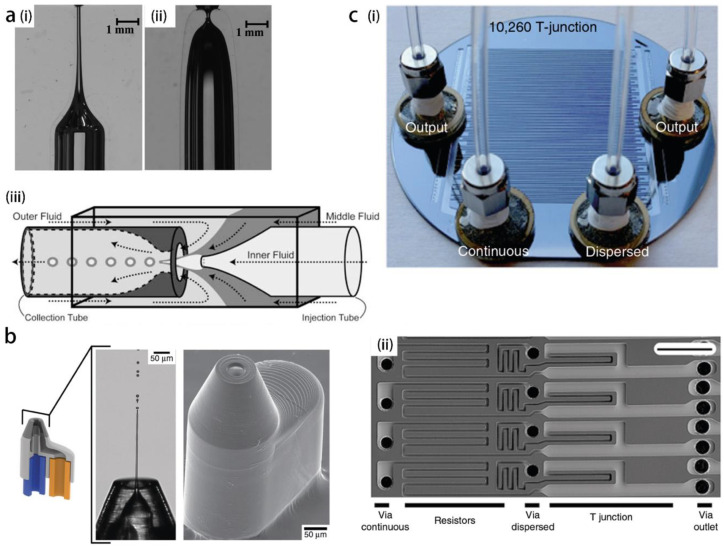
Recently developed strategy for micro-device fabrication. (**a**) A comparison between pulled capillary (**i**) and flame-shaped capillary (**ii**), and the application of flame-shaped capillary to be assembled with other capillaries for core-shell droplet generation (**iii**). Reproduced from [125,126], copyright 2005, AAAS and copyright 2018, Elsevier. (**b**) 3D micro-nozzle printed with direct laser writing, that can easily be assembled with feeding capillaries to form submicron jet at speed exceeding 160 m/s. Reproduced from [129], copyright 2020, Springer Nature. (**c**) Silicon-glass micro-device with 10,260 parallelized channels for micro-droplet generation, the whole view of the device (**i**) and zoom on the channels (**ii**), feeding channels for fluids are etched on the back side of the silicon wafer, opposed to the droplet generation channels. Reproduced from [100], copyright 2018, Springer Nature.

## 5. Conclusions

The microfluidic method was widely considered to be advantageous in generating highly monodisperse micron-sized or bigger droplets because of the precise manipulation of fluid. The droplets can be used as containers for biomedical analysis and chemical reactions, and can also be used as carriers of functional components for medical treatment or oil recovery. Due to the rising demand of exploring the highly confined pathways down to submicron scale, such as tissue interstitials or rock fissures, droplet size is expected to reach submicron scale, while maintaining monodispersity, which is beneficial for stability and functional performance. The production rate is also expected to rise in order to perform higher throughput analysis, such as for the DNA sequencing [133]. The channel fabrication resolution is the principal limiting factor for generating submicron droplets in most of the microfluidic geometries. However, the tipstreaming method including the hydrodynamic tipstreaming and the electrospray are not significantly prone to size of the orifice, and the step-emulsification method relies mostly on channel depth, which is technically less challenging to control than channel width in the nanofabrication process. The works mentioned in this review have achieved or are promising in the generation of submicron droplets using microfluidic channels, with theoretical models established. With the development of technology for a rigid device with good replicability, such as high-resolution 3D printing, microfluidics may lead the way towards generating submicron droplets with improved production rate and reproducibility.

## Figures and Tables

**Figure 1 micromachines-14-00638-f001:**
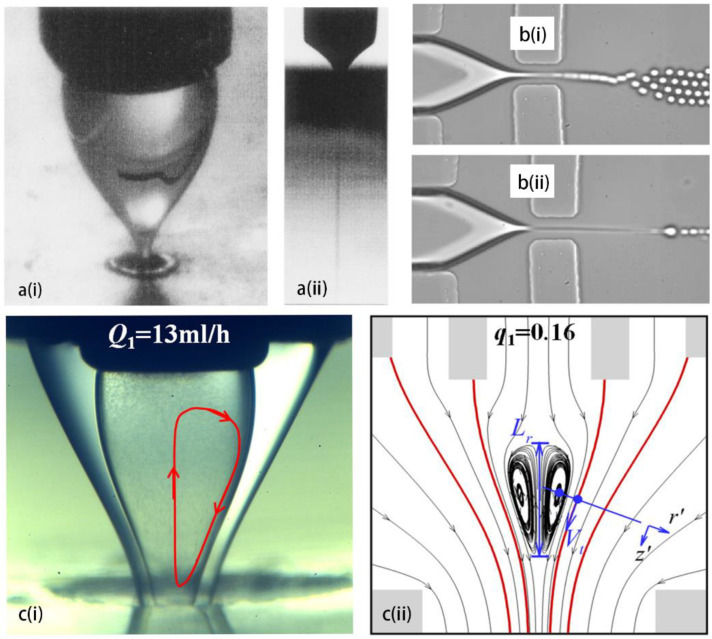
Methods of hydrodynamic tipstreaming. (**a**) 3D flow focusing of disperse fluid by gas stream through an orifice (**i**), and the formed thin jet which eventually breaks into tiny droplets (**ii**), reproduced from [29], copyright 1998, American Physical Society. (**b**) Tipstreaming in a planar PDMS channel from dripping mode (**i**) and jetting mode (**ii**), reproduced from [34], copyright 2003, AIP. (**c**) Conical interface of compound thread in co-axial flow focusing, recirculation zone in the cone is illustrated by PIV (**i**), and by numerical calculation (**ii**), reproduced from [36], copyright 2022, American Physical Society.

**Figure 2 micromachines-14-00638-f002:**
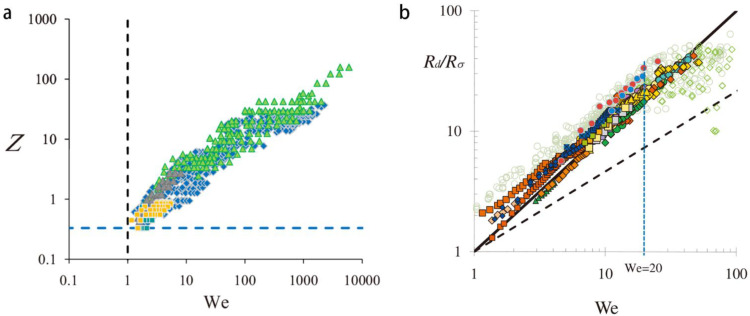
(**a**) Operational conditions of hydrodynamic flow focusing for having steady cone-jet structure in the plane of We and $Z=\tau_{s}D/\sigma$ (the ratio between tangential shear stress $\tau_{s}$ and surface tension stress, with D the orifice diameter and σ the surface tension). (**b**) Normalized droplet radius $R_{d}/R_{\sigma }$ vs. We for both flow focusing and electrospray. The symbols signify different fluids. The solid line is consistent with Equation (1). Reproduced from [37], copyright 2009, American Physical Society.

**Figure 6 micromachines-14-00638-f006:**
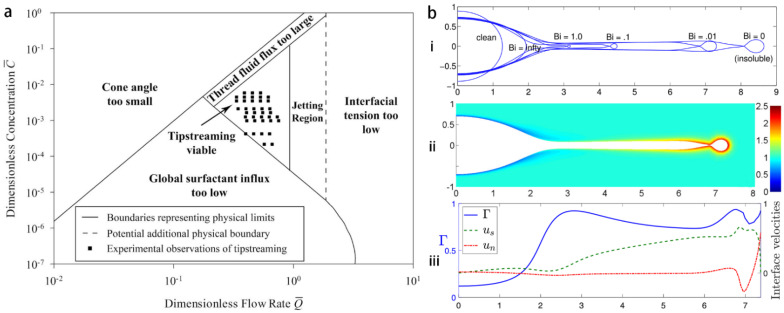
(**a**) Comparison between predicted operational condition of tipstreaming in planar device with the experimental data, in plane of dimensionless surfactant concentration and dimensionless flow rate of disperse fluid. The theoretical boundaries account for the interfacial mass balance of surfactants, global mass balance of fluid, geometry, transition to jetting, and minimum value of interfacial tension. Reproduced from [71], copyright 2012, AIP. (**b**) The length of the jet at first pinch for different Biot numbers (**i**), distribution of the surfactant concentration (**ii**), and an abrupt increase in surfactant concentration is observed at the interface of the jet downstream of the cone (blue line in (**iii**)). Reproduced from [72], copyright 2018, American Physical Society.

**Figure 7 micromachines-14-00638-f007:**
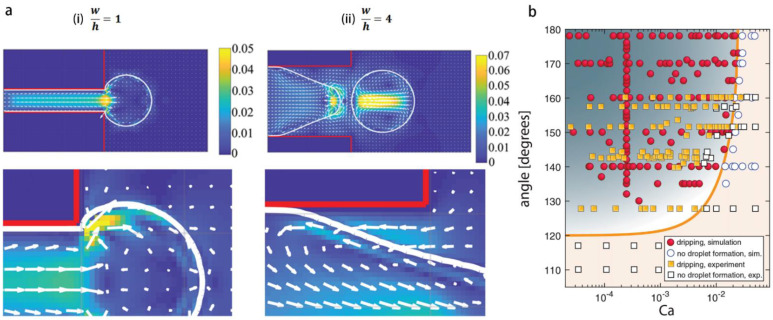
Operational conditions for step-emulsification of highly monodisperse droplets. (**a**) Flow field in channel with aspect ratio 1 (**i**) and 4 (**ii**). A back flow is allowed in the channel with aspect ratio 4, which promotes pinching at the step. Reproduced from [81], copyright 2019, AIP. (**b**) Phase diagram of droplet formation in plane of capillary number and contact angle of the disperse fluid on channel wall. A minimum 120° is required for having dripping mode, which generates monodisperse droplets at high rate. Reproduced from [82], copyright 2018, NAS of the USA.

**Figure 8 micromachines-14-00638-f008:**
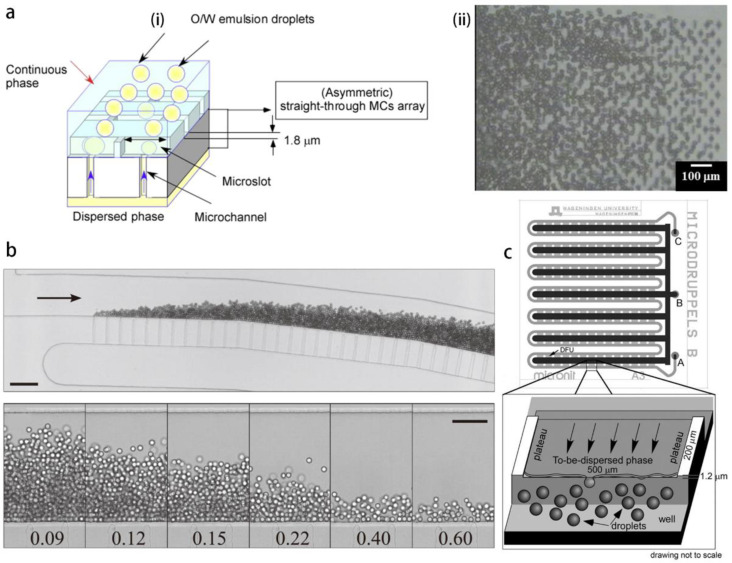
Principal step-emulsification strategies for enhancing droplet production rate. (**a**) Membrane with asymmetric straight-through microchannels (**i**) and a whole view of droplet generation (**ii**). Reproduced from [88], copyright 2017, WILEY-VCH Verlag GmbH & Co. KGaA, Weinheim. (**b**) Emulsification via parallelized channels to form microdroplets. Reproduced from [77], copyright 2014, AIP. (**c**) Edge-based droplet generator composed of a number of ultra-shallow channels, that each of them forms microdroplets along different spanwise positions. Reproduced from [92], copyright 2010, Elsevier.

**Figure 9 micromachines-14-00638-f009:**
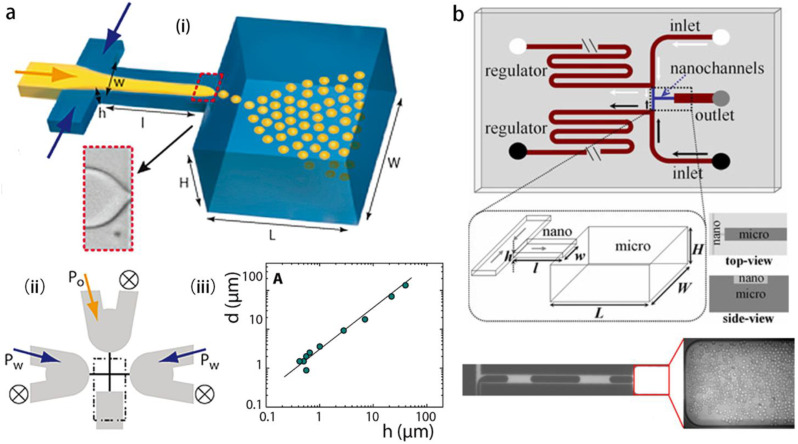
Co-flow step-emulsification method that generates micron and submicron droplets by downsizing the channel depth. (**a**) Device with shallow channel connected with a wide and deep reservoir (**i**), wide and deep “river” channels that prevent clogging at nanochannel (**ii**), and droplet diameter vs. channel depth (**iii**). Reproduced from [98], copyright 2010, American Chemical Society. (**b**) Submicron droplets formed in the step-emulsification nano-device, reproduced from [99], copyright 2011, Springer.

**Table 1 micromachines-14-00638-t001:** Comparison of the methods for generating submicron droplets. (*) signifies data not explicitly presented in the reference, but are inferred by the authors based on the provided information.

Reference	Category	Method	Minimum Size (Polydispersity) or Size Range	Production Rate	Compositions
[23]	Bulk method	high pressure homogenization and ultrasonication.	80 nm–350 nm (CV = 20%)	~$4\times 10^{9}$ droplets/s *	Silicone oil in SDS aqueous solution
[26]		Bubble bursting	59.8 nm (PDI = 0.091)	N/A	Hexadecane in CTAB aqueous solution
[106]		spontaneous emulsification	163 ± 2 nm	N/A	α- tocopherol and acetone
[34]	Hydrodynamic tipstreaming and electrospray	tipstreaming (flow focusing)	~O(100) nm	~$1.6\times 10^{7}$ droplets/s *	Water in oil
[42]		tipstreaming (3D flow focusing)	~200 nm (PDI < 0.04)	>$10^{6}$ droplets/s *	PFC in glycerol solution
[43]		atomization in PDMS channel	1–8 μm	$10^{8}–10^{9}$ bubbles/mL	$C_{4}F_{10}$ in lipid solution
[45]		opposed flow focusing	~1 μm	N/A	Silicone oil in glycerol solution
[73]		tipstreaming in planar PDMS channel	1–2 μm (CV < 2%)	N/A	Water in oil
[107]		tipstreaming (co-flow)	~1 μm (CV < 10% *)	~$10^{4}$ droplets/s	Silicone oil in glycerine
[108]		tipstreaming (flow focusing by gas)	~500 nm	~$5\times 10^{6}$ droplets/s	Water in helium gas
[62]		electrospray in PDMS channel	<3 μm	~3500 droplets/s	alginate Janus and PLGA Janus in mineral oil
[109]		electrospray	70 nm	N/A	Liquid crystal in air
[88]	Step-emulsification	microchannel emulsification	<5 μm	$2\times 10^{6}–2\times 10^{7}$ droplets/s *	Soybean oil in aqueous solution
[93]		EDGE emulsification	9 μm (CV < 5%)	$3.5\times 10^{4}$ droplets/s	Hexadecane in SDS aqueous solution
[98]		step-emulsification	0.9 μm (CV = 1%)	$1.5\times 10^{4}$ droplets/s	Fluorinated oil in water
[99]		step-emulsification	0.4 μm	~$10^{4}$ droplets/s	Hexadecane in water with fluorescein sodium salt
[101]	Microfluidics combined with low-energy methods	flow focusing and phase condensation	370 nm (CV 0.002–0.04%)	$6.5\times 10^{7}$ droplets/s *	$C_{5}F_{12}$ in glycerol aqueous solution
[102]		flow focusing and solvent dissolution	470 nm (CV = 3.8%)	$3\times 10^{6}$ droplets/s	PFC and DEE in water
[103]		flow focusing and evaporation	1.3 μm (CV = 15.3% *)	368 droplets/s *	Mixture of high- and low-saturate-vapor-pressure fluids in FC40 oil
[104]		step-emulsification and gas diffusion	700 nm (CV = 8%)	N/A	${\mathrm{CO}}_{2}$ and $C_{6}F_{14}$ in glycerol solution
[67]	Porous media made of PDMS	Microfiber-templated microfluidic chip	45.7 nm–1.03 μm (CV < 20%)	1.63$\times 10^{10}$ droplets/s	PEGDA in mineral oil

## Data Availability

Data can be provided upon reasonable request.

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
