# Peer review of "Microfluidic Methods for Generation of Submicron Droplets: A Review"

_micromachines, 2023, doi:10.3390/mi14030638_

Round 1

Reviewer 1 Report

The paper covers an interesting and promising topic of fabrication of submicron droplets using microfluidics. The paper has a good presentation, however it requires minor changes before publication.

1) The title of the paper is confusing because the reviewed methods can provide only submicron droplets. I recommend to change in the title nanodroplets to submicron droplets.

2) Please, make a comparison table with characteristics of all reviewed methods. 

3) on Fig 3c the droplets diameters distribution is rather wide. Please discuss and indicate in the comparison table how wide is the droplets diameters distribution produced by all reviewed methods. 

4) The images quality should be improved. Also I recommend to make the font larger on them. For example on fig.3 its hard to read all the notes on the images.

5) Panels of figures 4 and 5  should be rearranged according to the way how they are cited in the main text. Currently fig 4a and then 5a is cited.

6) The authors write that by electrospray core-shell and janus particles can be produced. But its unclear how it is technically organized. I recommend to add an illustration of such particles on fig 4.

 7) From Fig. 4c it's hard to get the idea what is on it. I recommend to revise this figure and its capture.  

Reviewer 2 Report

This review introduces the methods for generating nanodroplets, focusing on some microfluidic methods and comparing them with other methods. The article also summarizes some emerging applications of nanodroplets, as well as the device fabrication. This paper is well written and organized, and it can be published after addressing some minor issues.

(1) The use of terminologies such as nanodroplets, macroemulsions, submicron droplets, and emulsions in the Introduction section is too confusing and inconsistent. It is recommended to use consistent terminology for the same object.

(2) In the section of "Bulk methods", it is better to directly put the figures from the references in your figures instead of using citations like “Fig. 5 in [22].

(3) I would like to give a table to clearly compare the advantages and disadvantages of different methods for nanodroplets generation.

Reviewer 3 Report

The manuscript entitled "Microfluidic methods for generation of nanodroplets: A review" reviewed the recent progress on microfluidics-based platforms for nanodroplets generation. Overall, it is well organized but needs some necessary revisions.

1. A comparison among different nanodroplet methods (bulk methods, microfluidic methods, microfluidics combined with low-energy methods) is needed to the best of readers.

2. The future and challenges of the microfluidics-based platforms for nanodroplets generation are not summarized quite well. It is better to discuss the current scenario, challenges and future from at least the perspective of device fabrication and throughput.

3. It is strange that there is no table in the current manuscript. For different nanodroplet methods, the authors are suggested to include tables for performance comparation.

4. For all figures, a consistent format should be used. The figures present in the current manuscript are very rough.

5. Abstract needs to be condensed.

Round 2

Reviewer 3 Report

Happy to see that the authors addressed all my comments.